# The Aerogen^®^ Solo Is an Alternative to the Small Particle Aerosol Generator (SPAG-2) for Administration of Inhaled Ribavirin

**DOI:** 10.3390/pharmaceutics12121163

**Published:** 2020-11-29

**Authors:** Ronald H. Dallas, Jason K. Rains, Keith Wilder, William Humphrey, Shane J. Cross, Saad Ghafoor, Jessica N. Brazelton de Cardenas, Randall T. Hayden, Diego R. Hijano

**Affiliations:** 1Department of Infectious Diseases, St. Jude Children Research Hospital, Memphis, TN 38105, USA; ronald.dallas@stjude.org; 2Department of Critical Care and Pulmonary Medicine, St. Jude Children Research Hospital, Memphis, TN 38105, USA; jason.rains@stjude.org (J.K.R.); keith.wilder@stjude.org (K.W.); saad.ghafoor@stjude.org (S.G.); 3Department of Pharmaceutical Sciences, St. Jude Children Research Hospital, Memphis, TN 38105, USA; william.humphrey@stjude.org (W.H.); shane.cross@stjude.org (S.J.C.); 4Department of Pathology, St. Jude Children Research Hospital, Memphis, TN 38105, USA; jessica.brazeltondecardenas@stjude.org (J.N.B.d.C.); randall.hayden@stjude.org (R.T.H.)

**Keywords:** RSV, ribavirin, Aerogen^®^ Solo, SPAG-2, children, infection

## Abstract

Respiratory syncytial virus (RSV) is associated with adverse outcomes among immunocompromised patients. Inhaled ribavirin has been shown to improve mortality rates. The Small-Particle Aerosol Generator delivery system (SPAG-2) is the only FDA-cleared device to deliver inhaled ribavirin. However, it is difficult to set up and maintain. We developed a method for delivery of this medication using the vibrating mesh nebulizer (Aerogen^®^). We did not observe any adverse events with this method.

## 1. Introduction

Respiratory syncytial virus (RSV) is the leading cause of viral pneumonia in children. Infants born prematurely or with cardiac disease as well as severely immunocompromised patients are at increased risk for morbidity and mortality related to RSV infection [1,2]. Poor outcomes such as progression to pneumonia and respiratory failure occur more frequently among immunocompromised patients when compared those who are immunocompetent [3]. Progression to pneumonia has been described in up to half of adults with cancer. Mortality for those who developed lower respiratory tract infection (LRTI) has been reported to be as high as 75% [4]. Among children undergoing hematopoietic cell transplant (HCT), mortality associated with respiratory viruses usually ranges between 10% and 14% but can be as high as 30% [5].

Antiviral treatment with inhaled ribavirin has been shown to improve mortality rates compared to no treatment [6]. The Small-Particle Aerosol Generator delivery system (SPAG-2) is the only Food and Drug Administration (FDA) cleared device to deliver ribavirin [7]. The SPAG-2 is a pneumatically powered nebulizer designed specifically for the administration of ribavirin and therefore uncommon in respiratory therapy for delivering aerosolized medications [8]. In addition, setup of the machine is time consuming and difficult to maintain. The entire unit has to be disassembled and cleaned between each individual treatment, thus decreasing availability if multiple patients require inhaled ribavirin simultaneously. Furthermore, during the 2015–2016 and 2017–2018 winter seasons, there were two FDA recalls of certain SPAG-2 units threatening the access of this treatment to our patients [9,10]. Subsequently, at St. Jude Children’s Research Hospital, we developed and implemented an alternative method to deliver inhaled ribavirin using the Aerogen^®^ Solo vibrating mesh nebulizer. This nebulizer, which is a self-contained device, can be reused after flushing and cleaning with antimicrobial wipes. It is also smaller and easier to store in biohazard bags between uses. Finally, unlike the SPAG-2 device, it allows administration of inhaled ribavirin during periods of mechanical ventilation. In this article, we describe the development and implementation of the Aerogen^®^ Solo vibrating mesh nebulizer at our institution for inhaled ribavirin delivery.

## 2. Materials and Methods

During the 2017–2018 winter RSV season, the SPAG-2 devices were recalled. As a result, we developed a system to deliver inhaled ribavirin with the Aerogen^®^ Solo so we could continue offering ribavirin to patients with proven RSV infection. Indication and duration for inhaled ribavirin treatment was performed per institutional guidelines [11]. A total of 12 patients received inhaled ribavirin via this new method. In a subset of 5 patients with leftover respiratory samples available for testing, RSV loads were measured before and after ribavirin treatment using quantitative ddPCR [12].

Delivery of inhaled ribavirin via the Aerogen^®^ Solo vibrating mesh nebulizer included the use of a MedFusion 3500 syringe pump (Smiths Medical), a demistifier 2000 HEPA filtration system and a demistifier negative pressure bed enclosure with canopy around the patient’s bed (Peace Medical Inc., Wharton, NJ, USA). 

Two grams of ribavirin (Virazole; Bausch Health, Laval, QC, Canada) reconstituted in 30 mL of normal saline were administered over 2 h 14 min at an oxygen flow rate of 8 L/min and a pump rate of 14 mL/h. (Figure 1). Initially, the oxygen flow rate was 4–6 L/min, but crystal formation in the nebulizer chamber was observed, causing concern for blockage/obstruction. To correct this, ribavirin was further diluted in 100 mL of normal saline; however, this caused spillage of ribavirin at pump rates above 14 mL/h. The oxygen flow rate was then modified to 8 L/min, allowing for safe administration of 2 g ribavirin diluted in 30 mL of saline over two hours with a pump rate of 12–14 mL/h without observing crystallization and/or spillage. In addition, there was less wasted ribavirin in the nebulizer chamber using the Aerogen^®^ Solo when compared to SPAG by estimation of respiratory therapy staff. Finally, we attempted dosing 6 g of ribavirin over 12–18 h and again noted issues with drug crystallization in the nebulizer chamber.

At the end of each prescribed treatment, the Aerogen^®^ nebulizer was turned off. The demistifier and any drug remaining in the nebulizer or IV tubing were placed into a red biohazard bag along with the large bore tubing, T adaptor, aerosol mask and O2 reduce, and disposed according to institutional policy. While vibrating-mesh nebulizer has been re-used for 28 days with albuterol, given the potential for the crystals to block the membrane the nebulizer was only used for 24 h and then replaced with a new device [13]. Upon completion of the last prescribed treatment, the same process was followed including the demistifier canopy, aerosol tubing and mask. The demistifier, bed enclosure and syringe pump were cleaned with antimicrobial wipes prior to storage. After this procedure, environmental health services proceeded with decontamination of the inpatient hospital room (Appendix A).

The study was approved by the St Jude Children’s Research Hospital Institutional Review Board #00000029 (FWA00004775) on 21 September 2020 and determined to be secondary research and exempt under the Office for Human Subject Protection category 4, for which consent is not required (IRB number 20-0585).

## 3. Results

During the 2017–2018 and 2018–2019 RSV seasons, 12 patients were treated for RSV infection: 7 in 2017–2018 and 8 in 2018–2019 (Table 1). The median age was 9.5 years (range 5–19 years), and 5 were male (41.66%). All patients had leukemia with acute lymphoblastic leukemia (ALL) being the most frequent diagnosis (*n* = 8), followed by acute myeloid leukemia (AML; *n* = 4). Six patients previously underwent HCT, and 2 patients were undergoing HCT conditioning at the time of RSV diagnosis. Ten patients (66.7%) presented with upper respiratory tract infection (URTI), two of which progressed to LRTI. The remaining two patients (16.6%) presented with LRTI at the time of diagnosis. Eight patients (66.6%) had profound lymphopenia (absolute lymphocyte count < 200 cells/μL) at the time of diagnosis. The duration of ribavirin therapy ranged from 2 to 10 days. According to institutional guidelines, patients with URTI received 5 days of inhaled ribavirin and patients with LRTI received 10 days. Three patients had chronic lung disease, 2 of them presenting with or developing LRTI.

No patients received palivizumab as prophylaxis, nor did they receive it as part of their treatment regimen. During the 2018–2019 season, we began using oral (PO) ribavirin in select cases and three patients (cases 6, 9 and 10) were transitioned to PO ribavirin.

In a subgroup of 5 patients, we were able to measure viral loads from leftover respiratory samples before and after treatment. We observed a reduction of 3.92 log copies/mL of RSV viral load in respiratory secretions upon completion of antiviral therapy. We did not observe any bronchospasm associated with administration of ribavirin using either device.

## 4. Discussion

Management of RSV infection in immunocompromised patients remains a challenge due to its high morbidity and mortality. Supportive treatment and inhaled ribavirin are the only available options in this setting. The latter, while widely used, is expensive, has significant side effects and has been difficult to administer due to complications with the SPAG-2. Here, we describe an alternative method for administering inhaled ribavirin with the Aerogen^®^ Solo mesh nebulizer.

Ribavirin is a nucleoside analog with broad activity against DNA and RNA viruses that can be administered by inhalation, oral or intravenous routes (the latter only available through investigational new drug application and approval). Inhaled ribavirin can be administered by the SPGA-2 device in two dosing approaches: 2 g diluted in 30 mL of normal saline over 2 h, three times daily; or 6 g diluted on 100 mL over 12–18 h per day. Although there are no clinical trials comparing these regimens, there does not seem to be any clinical benefits of either regimen based on the current literature [14,15,16,17]. Inhaled ribavirin in either approach decreases progression to LRTI and mortality in immunocompromised patients. There is a dearth of evidence regarding the efficacy of oral or intravenous ribavirin for the treatment of RSV infection. The former has poor oral bioavailability and the latter carries a greater risk of toxicity and is difficult to access [18].

There is very little information on the performance of these devices while administering inhaled ribavirin. Walsh et al. performed an invitro study using 2 models of delivery (mechanical ventilation and mask) to determine the effects of the vibrating mesh micropump technology on the concentration of ribavirin and to characterize ribavirin delivery by particle size distribution and median aerodynamic diameter comparing the SPAG-2 and a vibrating mesh micropump. Although the small-particle aerosol generator produced a smaller mass median aerodynamic diameter than the vibrating mesh micropump, there was no significant difference in the proportion of drug mass in and the total drug delivery was similar among both devices [19]. Studies like this are especially important, given that clinical studies comparing both devices would be limited by the small number of patients that would be eligible to receive inhaled ribavirin and unlikely to be feasible.

Our results are limited to the small number of cases and retrospective nature of the design. In addition, leftover samples to measure viral load were only available for a subset of patients. Finally, we used historical data to compare both devices since the SPAG-2 became unavailable due to the recall. For all these reason, safety and efficacy could not be compared. During the transition to the Aerogen^®^ Solo, all treatments were carefully monitored by the respiratory therapist. We will continue to monitor side effects and clinical recovery as we have widely adopted this method in our institution. Furthermore, this method has the potential to be used in patients requiring mechanical ventilation, something that is complicated and challenging with the SPAG-2 device. Although we have not used it for this specific situation, a detailed process is available in our institutional standard operating procedure if the need were to arise (Appendix A).

In conclusion, we present an alternative to administer inhaled ribavirin with a widely used nebulizer, The Aerogen^®^ Solo, that can simplify the process with no additional side effects or changes in clinical outcomes.

## Figures and Tables

**Figure 1 pharmaceutics-12-01163-f001:**
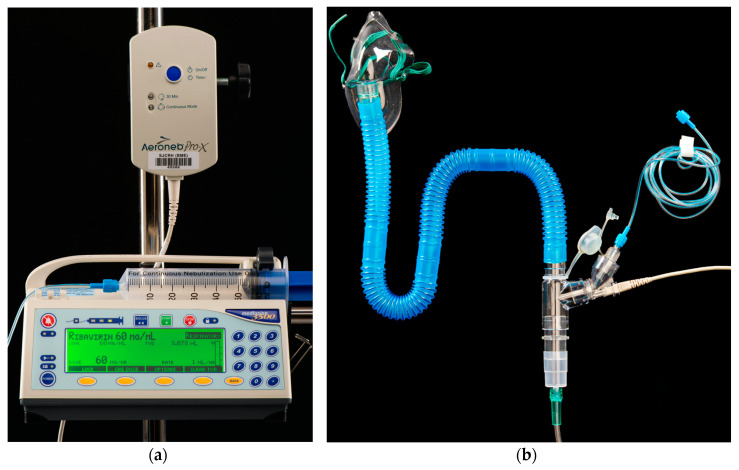
Set up and connection to administer inhaled ribavirin with the Aerogen^®^ Solo (**a**) MedFusion 3500 syringe pump containing ribavirin at a concentration of 60 mg/mL connected to nebulizer; (**b**) aerosol mask, large bore tubing and T adaptor connection.

**Table 1 pharmaceutics-12-01163-t001:** Clinical characteristics of patients with respiratory syncytial virus (RSV) infection treated with inhaled ribavirin.

Case	Age (Years)	Gender	Diagnosis	URTI/LRTI	ALC	Days Therapy	Transition to PO	Comments
1	13	F	ALL (s/p haplo-HCT + 28)	URTI	75	5	No	
2	14	F	ALL (s/p MUD-HCT + 83)	URTI	93	5	No	Chronic lung disease
3	7	F	AML(pre-HCT)	URTI→LRTI	0	10	No	
4	9	M	Relapsed ALL	URTI	0	5	No	
5	19	F	AML (s/p MUD-HCT >1 year)	URTI→LRTI	1480	7	No	Chronic lung disease
6	7	F	ALL (s/p haplo- HCT >1 year)	LRTI	1186	10	Yes	Chronic lung disease
7	9	M	ALL (s/p haplo-HCT + 16)	URTI	0	5	No	
8	10	M	ALL (s/p haplo-HCT + 132)	URTI	1630	5	No (initially started on PO)	
9	15	F	AML	LRTI	0	8	Yes	
10	6	M	ALL relapsed	URTI	468	5	Yes	
11	5	M	AML	URTI	0	2	NA	
12	11	F	ALL (pre-HCT)	URTI	0	10	NA	

M (male); F (female); acute myeloid leukemia (AML); acute lymphoblastic leukemia (ALL); hematopoietic cell transplant (HCT); match unrelated donor (MUD); lower respiratory tract infection (LRTI); upper respiratory tract infection (URTI); absolute lymphocyte count (ALC); small particle aerosol generator (SPAG-2).

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
