# Peer review of "The Aerogen® Solo Is an Alternative to the Small Particle Aerosol Generator (SPAG-2) for Administration of Inhaled Ribavirin"

_pharmaceutics, 2020, doi:10.3390/pharmaceutics12121163_

Round 1
Reviewer 1 Report
1) Lines 44-45 the authors describe the possibility to reuse the device after flushing and cleaning, however, in the paragraph between lines 58-70 they describe a phenomenon of crystallization which could cause harm to the membrane of the nebulizer by blocking its holes. Re-use of such device after ribavirin administration should be discouraged because it is not possible to inspect the membrane carefully. Please discuss this topic and implement in the text.
2) Material and methods. Please provide information if any on the dose calculation. The SPAG-2 relies on a technology that differs substantially from the Aerogen Solo. There is no information on the aerosol particle size distribution (APSD) of ribavirin with the Aerogen Solo, therefore the nature of the aerosol is unpredictable as well as the potential dose delivered to the lung. In particular in light of the crystallization observed this raises a significant concern with regards to the drug dose that could have been delivered to the lungs. This important study limitation should be acknowledged and included in the discussion highlighting the importance of in vitro studies to characterize the nebulization of ribavirin by means of the Aerogen Solo
Author Response
Response to Reviewer 1 Comments
Point 1: Lines 44-45 the authors describe the possibility to reuse the device after flushing and cleaning, however, in the paragraph between lines 58-70 they describe a phenomenon of crystallization which could cause harm to the membrane of the nebulizer by blocking its holes. Re-use of such device after ribavirin administration should be discouraged because it is not possible to inspect the membrane carefully. Please discuss this topic and implement in the text.
Response 1: Reviewer brings a good point. The crystallization mentioned in lines 58-70 points to crystals in the T adaptor, not crystals blocking the membrane of the nebulizer itself. However, we agree that these crystals could potentially block the membrane and for that reason we only re-use the device three times which account for a 24 hs period, as we give ribavirin TID, and the discard and use a new one the day after. We have clarified this in the manuscript.
Point 2: Material and methods. Please provide information if any on the dose calculation. The SPAG-2 relies on a technology that differs substantially from the Aerogen Solo. There is no information on the aerosol particle size distribution (APSD) of ribavirin with the Aerogen Solo, therefore the nature of the aerosol is unpredictable as well as the potential dose delivered to the lung. In particular in light of the crystallization observed this raises a significant concern with regards to the drug dose that could have been delivered to the lungs. This important study limitation should be acknowledged and included in the discussion highlighting the importance of in vitro studies to characterize the nebulization of ribavirin by means of the Aerogen Solo
Response 2: We appreciate the reviewer’s comment. Walsh et al, performed an invitro study using 2 models of delivery (mechanical ventilation and mask) to determine the effects of the vibrating mesh micropump technology on the concentration of ribavirin and to characterize ribavirin delivery by particle size distribution and median aerodynamic diameter comparing the SPAG-2 and a vibrating mesh micropump. They did not find a significant difference in ribavirin concentration before and after vibrating mesh micropump. They calculated the particle size distribution between devices and concluded that although the small-particle aerosol generator produces a smaller mass median aerodynamic diameter than the vibrating mesh micropump the drug mass in the respirable range was similar. In addition, the crystallization observed was lees that what we have seen over the past years when using the SPAG-2 unit. This along with the following reference have been added to the text:
Walsh BK, Betit P, Fink JB, Pereira LM, Arnold J. Characterization of Ribavirin Aerosol With Small Particle Aerosol Generator and Vibrating Mesh Micropump Aerosol Technologies. Respir Care. 2016 May;61(5):577-85. doi: 10.4187/respcare.04383. Epub 2016 Mar 1. PMID: 26932383.
Reviewer 2 Report
The paper under review deals with the research on the inhalation. In my opinion the subject is interesting but more scientific details are needed (now it looks like a report, not a research paper). The authors developed a method for drug delivery using a vibrating mesh nebulizer, but in my opinion there is not enough detail / data (e.g. what droplet diameters are obtained). The article contains only 16 literature items and 1 figure. The conclusions are needed. In opinion of the reviewer the article can be accepted for publication after major correction.
1)The experimental part of the article has to be revised. Please list the chemical and instruments producers.
2)Please provide information on the dose calculation.
3)There is no information on the spray particle size distribution and mean diameters. What droplet diameters are obtained? It is crucial to describe the phenomenon.
4)If possible, please add a plot of particle size distribution and any data on mean droplet diameter.
5)Fig. 1b is too small – please add the geometry of the nebuliser.
Author Response
Response to Reviewer 2 Comments
Point 1:The experimental part of the article has to be revised. Please list the chemical and instruments producers.
Response 1: We have rewritten the paragraph to include the requested information as follows:
Delivery of inhaled ribavirin via the Aerogen® Solo vibrating mesh nebulizer included the use of a MedFusion 3500 syringe pump (Smiths Medical), a demistifier 2000 HEPA filtration system and a demistifier negative pressure bed enclosure with canopy around the patient’s bed (Peace Medical Inc). Two grams of ribavirin (Virazole; Bausch Health) reconstituted in 30 mL of normal saline were administered over 2 hours 14 minutes at an oxygen flow rate of 8 L/min and a pump rate of 14 mL/hr. (Figure 1).
Point 2: Please provide information on the dose calculation.
Response 2: Inhaled ribavirin can be administered daily via continuous infusion over a 12- to 18-hour period from a solution containing 20 mg/ml, or as high dose short intervals over two hours three times a day from a solution containing 60mg/ml. The following references are in the manuscript.
Waghmare, A., J.A. Englund, and M. Boeckh, How I treat respiratory viral infections in the setting of intensive chemotherapy or hematopoietic cell transplantation. Blood, 2016. 127(22): p. 2682-92.
Englund, J.A., et al., High-dose, short-duration ribavirin aerosol therapy compared with standard ribavirin therapy in children with suspected respiratory syncytial virus infection. J Pediatr, 1994. 125(4): p. 635-41.
Knight, V., et al., High dose-short duration ribavirin aerosol treatment--a review. Bull Int Union Tuberc Lung Dis, 1991. 66(2-3): p. 97-101.
Englund, J.A., et al., High-dose, short-duration ribavirin aerosol therapy in children with suspected respiratory syncytial virus infection. J Pediatr, 1990. 117(2 Pt 1): p. 313-20.
Chemaly, R.F., D.P. Shah, and M.J. Boeckh, Management of respiratory viral infections in hematopoietic cell transplant recipients and patients with hematologic malignancies. Clin Infect Dis, 2014. 59 Suppl 5: p. S344-51.
Point 3: There is no information on the spray particle size distribution and mean diameters. What droplet diameters are obtained? It is crucial to describe the phenomenon.
Response 3: Walsh et al, performed an invitro study using 2 models of delivery (mechanical ventilation and mask) to determine the effects of the vibrating mesh micropump technology on the concentration of ribavirin and to characterize ribavirin delivery by particle size distribution and median aerodynamic diameter comparing the SPAG-2 and a vibrating mesh micropump. The small-particle aerosol generator produced a smaller mass median aerodynamic diameter (1.84 mm) than the vibrating mesh micropump (3.63 mm); however, there was no significant difference in the proportion of drug mass in the 0.7– 4.7-mm particle range. Total drug delivery was similar with the small particle aerosol generator and vibrating mesh micropump in both spontaneously breathing and mechanical ventilation models. This along with the following reference have been added to the text: Walsh BK, Betit P, Fink JB, Pereira LM, Arnold J. Characterization of Ribavirin Aerosol With Small Particle Aerosol Generator and Vibrating Mesh Micropump Aerosol Technologies. Respir Care. 2016 May;61(5):577-85. doi: 10.4187/respcare.04383. Epub 2016 Mar 1. PMID: 26932383.
Point 4: If possible, please add a plot of particle size distribution and any data on mean droplet diameter.
Response 4: Unfortunately, that information is not available at this time. However, Walsh et al have described this in 2 in vitro models of delivery (mechanical ventilation and mask). See answer to Point 5 below.
Point 5: Fig. 1b is too small – please add the geometry of the nebuliser.
Response 5: We have updated the figure to make it clearer. Regarding the aerosol size comparison, Walsh et al, performed an invitro study using 2 models of delivery (mechanical ventilation and mask) to determine the effects of the vibrating mesh micropump technology on the concentration of ribavirin and to characterize ribavirin delivery by particle size distribution and median aerodynamic diameter comparing the SPAG-2 and a vibrating mesh micropump. This reference has been added to the text. Walsh BK, Betit P, Fink JB, Pereira LM, Arnold J. Characterization of Ribavirin Aerosol With Small Particle Aerosol Generator and Vibrating Mesh Micropump Aerosol Technologies. Respir Care. 2016 May;61(5):577-85. doi: 10.4187/respcare.04383. Epub 2016 Mar 1. PMID: 26932383.
Reviewer 3 Report
It is a well-attempted paper that explored and reported the clinical performance of the vibrating-mesh nebulizer Aerogen® Solo as an alterante approach for inhaled ribavarin administration.
The only major comment is:
- The manuscript only presented the results from Aerogen Solo related study. Since the title primarily focused on Aerogen Solo as an “alternative to” the SPAG2, it would be helpful to the readers to add/discuss any literature/historical study results about the SPAG2 device, and how Aerogen Solo performed compared to SPAG2. For example:
- If there are any literature reports on Aerosol particle size distribution generated by SPAG2 and (vs.) Aerogen Solo.
- The authors presented the viral load reduction of 3.92 log copies/ml after completion of the therapy by Aerogen Solo, how does the treatment with a similar drug dose using the SPAG2 device would yield the viral load reduction? (There is a recent article published in Pharmaceutics that characterized the in vitro performance of the Aerogen Solo device, any key supporting information relevant to this Manuscript study can be added. (Lin, Hui-Ling, et al. "In Vitro Evaluation of a Vibrating-Mesh Nebulizer Repeatedly Use over 28 Days." Pharmaceutics10 (2020): 971))
- The authors presented the cons of the SPAG2 system, what are the advantages of SPAG2 compared to Aerogen Solo?
Author Response
Response to Reviewer 3 Comments
Point 1: If there are any literature reports on Aerosol particle size distribution generated by SPAG2 and (vs.) Aerogen Solo.
Response 1: Walsh et al, performed an invitro study using 2 models of delivery (mechanical ventilation and mask) to determine the effects of the vibrating mesh micropump technology on the concentration of ribavirin and to characterize ribavirin delivery by particle size distribution and median aerodynamic diameter comparing the SPAG-2 and a vibrating mesh micropump. This reference has been added to the text. Walsh BK, Betit P, Fink JB, Pereira LM, Arnold J. Characterization of Ribavirin Aerosol With Small Particle Aerosol Generator and Vibrating Mesh Micropump Aerosol Technologies. Respir Care. 2016 May;61(5):577-85. doi: 10.4187/respcare.04383. Epub 2016 Mar 1. PMID: 26932383.
Point 2: The authors presented the viral load reduction of 3.92 log copies/ml after completion of the therapy by Aerogen Solo, how does the treatment with a similar drug dose using the SPAG2 device would yield the viral load reduction? (There is a recent article published in Pharmaceutics that characterized the in vitro performance of the Aerogen Solo device, any key supporting information relevant to this Manuscript study can be added. (Lin, Hui-Ling, et al. "In Vitro Evaluation of a Vibrating-Mesh Nebulizer Repeatedly Use over 28 Days." Pharmaceutics10 (2020): 971))
Response 2: Yes, we had observed a similar reduction in the viral load rate when using the SPAG-2 (2.48 log copies/ml) after completion of the therapy. The reference suggested by reviewer was added to the manuscript.
Point 3: The authors presented the cons of the SPAG2 system, what are the advantages of SPAG2 compared to Aerogen Solo?
Response 3: The main advantages of the SPAG-2 is that is FDA approved to deliver inhaled ribavirin, there is experience as it has been used for many years and allows for both methods, intermittent and continuous delivery, of inhaled ribavirin. We have included this in the discussion of the manuscript and as a supplementary table highlighting the pros and cons of each device:
SPAG-2 disadvantages observed at SJCRH
- Limited supply of SPAG available from manufacturer. SJCRH owned 10 complete SPAG originally. Pieces and parts broke and/or were lost in the sterilization process. Parts are unavailable for replacement from manufacturer.
- 2 recalls of SPAG-2 in past 5 years by FDA
- Device is large and unwieldy, requires separate table or stand, takes up space in crowded room.
- SPAG-2 set-up requires 6ft or more of large bore tubing and an aerosol mask. Hospital policy required that the disposable tubing and mask be removed after each treatment and placed into biohazard red bags to take into an incinerator.
- SPAG-2 had constant crystallization of large bore tubing requiring a minimum of Q1 tubing checks to ensure delivery of ribavirin to patient.
- Use of SPAG-2 at SJCRH resulted in pooled ribavirin crystals in sometimes large piles on the table with nebulizer, on the floor underneath the nebulizer and tubing, and in the patient bed.
- The SPAG requires the use of 7-10 liters of 100% oxygen to the patient.
- SPAG-2 demonstrated unreliable nebulization. It was observed that a treatment of a 300ml concentration of Ribavirin would consistently have 150ml or more drug remaining in the reservoir flask after 12 hours of continuous nebulization.
SPAG-2 advantages
- SPAG-2 is FDA approved for administration of aerosolized ribavirin.
- SPAG-2 allows multiple drug delivery concentrations and delivery times.
Aerogen Solo disadvantages observed at SJCRH
- The Aerogen Solo has a maximum nebulization rate of 12-14ml/hour which limits the dosage and concentration options for ribavirin delivery.
- The Aerogen Solo is prone to crystallization in the nebulizer chamber which can block all delivery of drug to the patient.
- The Aerogen Solo nebulizer cup often required a flush of sterile water between treatments to remove ribavirin residue which stops nebulization. (solved by increasing the driving flow from the flowmeter to >8L/min.
Aerogen Solo advantages
- Aerogen Solo nebulizer can be used multiple times without replacing the disposables.
- Aerogen Solo nebulizes the entire dose of Ribavirin and delivers to the patient.
- Aerogen Solo can deliver ribavirin to patient using oxygen or room air.
- Aerogen Solo leaves no crystallization on the floor or patient bed. Crystals remain in the short length of large bore tubing, the aerosol mask, and on the skin around the mouth and nose.
- Aerogen Solo can be contained to a single IV pole during and between treatments. The disposables may be placed in a labeled specimen zip lock bag and hung on the IV pole between use.
- Aerogen Solo parts are all disposable and replaceable.
Reviewer 4 Report
This communication depicted an off-label use of the mesh nebulizer Aerogen Solo as an alternative to the indicated SPAG-2 for the administration of inhaled ribavirin in twelve patients without any adverse events observed. While it is acknowledged the need of an alternative device and such communication should be beneficial to the community, I have the following comments for the authors' consideration.
Comments:
- The Materials and methods section has to be revised. Include company and manufacturer and origin of all chemicals (drugs) and instrument used. Particularly, the detailed settings (which appeared to be a history of optimization process) (line 58 to 65) should be made clearer. What were the reported flow rates corresponding to?
- The figure may be clearer if it can show the entire setup in one picture.
- Include in the discussion what further investigations can be done to systematically evaluate and compare the two devices, including but not limited to the in vitro evaluation of the aerodynamic particle size distribution of the aerosols generated by these devices.
- Elaborate and expand the limitations of the results. Any precautionary measures were in placed when device was changed?
Other comments:
- Spell out HCT at first appearance (line 31).
- Elaborate more why SPAG-2 is uncommon (line 35).
- Line 59: Is it "diluted" or reconstituted/dissolved?
- Line 96: Is it "dilution" or concentration?
- Proofread the first paragraph of discussion; it does not read smoothly.
Author Response
Response to Reviewer 4 Comments
Point 1: The Materials and methods section has to be revised. Include company and manufacturer and origin of all chemicals (drugs) and instrument used. Particularly, the detailed settings (which appeared to be a history of optimization process) (line 58 to 65) should be made clearer. What were the reported flow rates corresponding to?
Response 1: We have rewritten the paragraph to include the requested information and clearly distinguish oxygen flow rate and pump infusion rate as follows:
Delivery of inhaled ribavirin via the Aerogen® Solo vibrating mesh nebulizer included the use of a MedFusion 3500 syringe pump (Smiths Medical), a demistifier 2000 HEPA filtration system and a demistifier negative pressure bed enclosure with canopy around the patient’s bed (Peace Medical Inc). Two grams of ribavirin (Virazole; Bausch Health) reconstituted in 30 mL of normal saline were administered over 2 hours 14 minutes at an oxygen flow rate of 8 L/min and a pump rate of 14 mL/hr. (Figure 1). Initially, the oxygen flow rate was 4-6 L/min, but crystal formation in the nebulizer chamber was observed causing concern for blockage / obstruction. To correct this, ribavirin was further diluted in 100 mL of normal saline; however, this caused spillage of ribavirin at pump rates above 14 mL/hr. The oxygen flow rate was then modified to 8 L/min allowing for safe administration of 2 grams ribavirin diluted in 30 mL of saline over two hours with a pump rate of 12-14 mL/hr without observing crystallization and/or spillage.
Point 2: The figure may be clearer if it can show the entire setup in one picture.
Response 2: We understand and agree with reviewer. However, we do not have a photo with the whole set up available at this point.
Point 3: Include in the discussion what further investigations can be done to systematically evaluate and compare the two devices, including but not limited to the in vitro evaluation of the aerodynamic particle size distribution of the aerosols generated by these devices.
Response 3: Reviewer brings an important point. The following paragraph was added to the discussion.
There is very little information on the performance of these devices while administering inhaled ribavirin. Walsh et al, performed an invitro study using 2 models of delivery (mechanical ventilation and mask) to determine the effects of the vibrating mesh micropump technology on the concentration of ribavirin and to characterize ribavirin delivery by particle size distribution and median aerodynamic diameter comparing the SPAG-2 and a vibrating mesh micropump. Although the small-particle aerosol generator produced a smaller mass median aerodynamic diameter than the vibrating mesh micropump there was no significant difference in the proportion of drug mass in and the total drug delivery was similar among both devices [19]. Studies like this, are especially important, given that clinical studies comparing both devices would be limited by the small number of patients that would be eligible to receive inhaled ribavirin and unlikely to be feasible.
Walsh BK, Betit P, Fink JB, Pereira LM, Arnold J. Characterization of Ribavirin Aerosol With Small Particle Aerosol Generator and Vibrating Mesh Micropump Aerosol Technologies. Respir Care. 2016 May;61(5):577-85. doi: 10.4187/respcare.04383. Epub 2016 Mar 1. PMID: 26932383.
Point 4: Elaborate and expand the limitations of the results. Any precautionary measures were in placed when device was changed?
Response 4: The following paragraph was added to the discussion.
Our results are limited to the small number of cases and retrospective nature of the design. In addition, leftover samples to measure viral load was only available for a subset of patients. Finally, we used historical data to compare both devices as the SPAG-2 became unavailable during the recall. For all these reason, safety and efficacy could not be compared. During the transition to the Aerogen® Solo, all treatments were carefully monitored by the respiratory therapist. We will continue to monitor side effects and clinical recovery as we have widely adopted this method in the institution. Furthermore, this method has the potential to be used in patients requiring mechanical ventilation, something that is complicated and challenging with the SPAG-2 device. Although we have not used it in this situation, a detailed process is available in our institutional standard operating procedure if the need were to arise (supplementary material).
Point 5: Spell out HCT at first appearance (line 31).
Response 5: Corrected as hematopoietic cell transplant
Point 6: Elaborate more why SPAG-2 is uncommon (line 35).
Response 6: The following sentence and reference have been added.
The SPAG-2 is a pneumatically powered nebulizer designed specifically for the administration of ribavirin and therefore uncommon in respiratory therapy for delivering aerosolized medications.
White, G., Humidity and Aerosol Therapy Equipment, in Equipment Theory for Respiratory Care. 2014, Cengage Learning. p. 95-160.
Point 7: Line 59: Is it "diluted" or reconstituted/dissolved?
Response 7: This was corrected as reconstituted.
Point 8: Line 96: Is it "dilution" or concentration?
Response 8: This was corrected as concentration.
Point 9: Proofread the first paragraph of discussion; it does not read smoothly.
Response 9: The paragraph has been modified as follows:
Management of RSV infection in immunocompromised patients remains a challenge due to its high morbidity and mortality. Supportive treatment and inhaled ribavirin are the only available options in this setting. The latter, while widely used, is expensive, has significant side effects and has been difficult to administer due to complications with the SPAG-2. Here we describe an alternative method for administering inhaled ribavirin with the Aerogen® Solo mesh nebulizer.
Round 2
Reviewer 1 Report
the clarifications provided are sufficient
Reviewer 2 Report
The expected output is that I see the changed file and answers for my comments. I can't find the author's answers.
In my opinion the subject is interesting but more scientific details are needed. In opinion of the reviewer the article can be accepted for publication after major correction.
Comments (unchanged):
1)The experimental part of the article has to be revised. Please list
the chemical and instruments producers.
2)Please provide information on the dose calculation.
3)There is no information on the spray particle size distribution and
mean diameters. What droplet diameters are obtained? It is crucial to
describe the phenomenon.
4)If possible, please add a plot of particle size distribution and any
data on mean droplet diameter.
5)Fig. 1b is too small – please add the geometry of the nebuliser.
Reviewer 3 Report
Authors address my comments, and I do not have additional comments.